# A Confirmatory Factor Analysis of the Dementia Attitude Scale (DAS) in a European Case Series of Caregivers of People with Dementia Enrolled in the RECage Study

**DOI:** 10.3390/neurosci6020045

**Published:** 2025-05-20

**Authors:** Bruno Mario Cesana, Eleni Poptsi, Magda Tsolaki, Sverre Bergh, Andrea Fabbo, Lutz Frölich, Maria Cristina Jori, Carlo Alberto Defanti

**Affiliations:** 1Unit of Medical Statistics, Department of Clinical Sciences and Community Health, Biometry and Bioinformatics “Giulio A. Maccacaro” Faculty of Medicine and Surgery, University of Milan, 20122 Milan, Italy; brnmrcesana@gmail.com; 2Laboratory of Psychology, Section of Cognitive and Experimental Psychology, Faculty of Philosophy, School of Psychology, Aristotle University of Thessaloniki (AUTh), 54124 Thessaloniki, Greece; 3Greek Association of Alzheimer’s Disease and Related Disorders (GAADRD), 54643 Thessaloniki, Greece; tsolakim1@gmail.com; 41st Department of Neurology, School of Medicine, Aristotle University of Thessaloniki (AUTh), 54124 Thessaloniki, Greece; 5Research Centre for Age-Related Functional Decline and Disease, Innlandet Hospital Trust, 2313 Ottestad, Norway; sverre.bergh@sykehuset-innlandet.no; 6Geriatric Service-Cognitive Disorders and Dementia, Department of Primary Care, Local Health Authority of Modena (AUSL), 41124 Modena, Italy; a.fabbo@ausl.mo.it; 7Department of Geriatric Psychiatry, Central Institute of Mental Health, Medical Faculty Mannheim, Heidelberg University, 68159 Mannheim, Germany; lutz.froelich@zi-mannheim.de; 8Mediolanum Cardio Research, 20123 Milano, Italy; jori@mcr-med.com; 9FERB Alzheimer Centre, Gazzaniga, 24025 Bergamo, Italy; carloalberto.defanti@ferbonlus.com

**Keywords:** RECage study, DAS, confirmatory factor analysis, exploratory factor analysis

## Abstract

Background: The Dementia Attitude Scale (DAS) is a validated instrument used to capture the affective, behavioural, and cognitive components of attitudes toward people living with dementia (PLwD). This study conducts confirmatory and exploratory factor analyses (CFA and EFA) of the DAS assessed by caregivers of PLwD and BPSD enrolled in the RECage multicentre clinical trial. Methods: The baseline questionnaire was completed by 485 caregivers (29.7% male, 70.3% female), from six European countries, reflecting diverse cultural contexts. CFA tested the two-factor structure of the original model, while EFA thoroughly explored the factor structure. Results: The CFA results showed a poor model fit, with significant deviations from ideal values for RMSEA (0.0861), SRMSR (0.0781), and CFI (0.7117), showcasing an inadequate representation of the data. EFA revealed a three-factor structure, explaining the 45.2% variance for social comfort, 28.8% for social discomfort, and 25.9% for dementia knowledge. The social comfort items reflected positive caregiver attitudes toward PLwD, while social discomfort captured feelings of discomfort and uncertainty about caregiving. Dementia knowledge included items related to understanding dementia’s symptoms and needs. Conclusions: The three-factor model highlights the importance of emotional comfort, knowledge of dementia, and social discomfort as key dimensions in caregiver attitudes.

## 1. Introduction

The increasing prevalence of dementia worldwide is a pressing public health concern, with estimates indicating that approximately 50 million individuals are currently living with the condition, a number projected to exceed 150 million by 2050 if effective interventions are not implemented [1]. This growing number of people with dementia is accompanied by significant economic implications, with costs associated with dementia care expected to reach USD 2 trillion by 2030 [2,3]. The growing incidence of dementia requires a comprehensive understanding of the societal attitudes towards this condition, particularly the stigma that continues to affect communities globally.

Besides the high rates of AD, there is still a lack of appropriate education, as well as stigmatization of people with dementia and especially AD [4]. The stigma significantly impacts the lives of people with dementia, as well as their caregivers. Therefore, Alzheimer-related stigma is a multifaceted issue that requires healthcare professionals to attempt to change the negative beliefs about Alzheimer’s disease to avoid possible feelings of isolation and support the coordination of care for dementia patients [5]. Comprehensive strategies for stigma reduction can focus on education and awareness campaigns [6], community engagement initiatives, and support groups [7], whilst the promotion of positive narratives, through the sharing of personal stories and experiences, can humanize the condition and challenge stereotypes attributed to stigma [8]. These strategies may lead to a better understanding of neurocognitive diseases and more dementia-friendly societies.

Therefore, assessing attitudes toward dementia is crucial for understanding public perceptions and improving care practices. Several validated tools have been developed to evaluate these attitudes across different populations, including healthcare professionals, students, and the public. Specifically, some of the most commonly used scales comprise the following: (a) the Dementia Community Attitudes Questionnaire, which focuses on the stigma associated with dementia and the beliefs held by the public [9], (b) the Approaches to Dementia Questionnaire (ADQ) which assess recognition of personhood and hope regarding the capabilities of individuals with dementia [10], (c) the Dementia Knowledge Assessment Scale (DKAS) which provides insights into attitudes by evaluating understanding of dementia-related issues [11], (d) the General Practitioners Attitude and Confidence Scale for Dementia (GPACS-D) which assesses attitudes and confidence levels among primary care doctors regarding dementia management [12], and the (e) Alzheimer’s Disease Knowledge Scale (ADKS), measuresing knowledge about Alzheimer’s disease and its symptoms, which is closely linked to attitudes [13], and, finally, (f) the Dementia Attitude Scale (DAS), developed by O’Connor and McFadden in 2010 [14]. DAS is a validated instrument consisting of 20 items that captures the affective, behavioural, and cognitive components of attitudes toward individuals living with dementia.

The items are scored on a seven-point Likert scale ranging from “strongly disagree” to “strongly agree” [14]. The total score ranges from 20 to 140, with higher scores indicating more positive attitudes towards dementia [14,15]. Fourteen items have a positive polarity, so a higher score relates to more positive attitudes, and six items (2, 6, 8, 9, 16, and 17) have a negative polarity. Therefore, the scale’s total is calculated after having reversed the scoring of these “negative” items: 2, 6, 8, 9, 16, and 17. This scale is particularly relevant as it not only measures specific attitudes regarding dementia but also provides insights into the underlying beliefs and perceptions that influence how individuals interact with and care for people with dementia [16].

In the original validation study of O’Connor and McFadden, the exploratory factor analysis (EFA) found a two-factor structure and a high internal consistency with Cronbach’s alpha values exceeding 0.8 [14]. The final 20 items of the DAS were loaded onto two factors, defined as the (a) “Dementia Comfort” and (b) “Social Knowledge” factors [14]. The first factor, “Dementia Comfort”, expresses how people feel in the presence of someone diagnosed with dementia and comprises the following items: (1) It is rewarding to work with people who have ADRD (Alzheimer’s disease and related dementias); (2) I am afraid of people with ADRD; (4) I feel confident around people with ADRD; (5) I am comfortable touching people with ADRD; (6) I feel uncomfortable being around people with ADRD; (8) I am not very familiar with ADRD; (9) I would avoid an agitated person with ADRD; (13) I feel relaxed around people with ADRD; (16) I feel frustrated because I do not know how to help people with ADRD; and (17) I cannot imagine taking care of someone with ADRD.

The second factor (dementia knowledge) includes participants’ beliefs regarding the capabilities of individuals with dementia and comprises the following items: (3) People with ADRD can be creative; (7) Every person with Alzheimer’s Disease and Related Dementias (ADRD) has different needs; (10) People with ADRD like having familiar things nearby; (11) It is important to know the history of people with ADRD; (12) It is possible to enjoy interacting with people with ADRD; (14) People with ADRD can enjoy life; (15) People with ADRD can feel when others are kind to them; (18) I admire the coping skills of people with ADRD; (19) We can do a lot now to improve the lives of people with ADRD; and (20) Difficult behaviours may be a form of communication for people with ADRD. This model has been confirmed utilizing confirmatory factor analysis (CFA) on a different sample of 157 college students [14], allowing for a better understanding of attitudes and highlighting the importance of both knowledge and emotional comfort in shaping interactions with people living with dementia [15,16].

Although DAS is widely utilized in studies that attempt to assess attitudes about dementia and evaluate the effectiveness of psychoeducation programs or other strategies for coping with the associated stigma observed in dementia, there are not many validation studies that investigate its psychometric properties. Nonetheless, the limited number of these studies indicates that the DAS maintains its factor structure across diverse groups, including healthcare professionals and students [17,18].

Further factor analyses of the DAS have been conducted to establish its construct validity and reliability across various populations. For instance, a study by Leung et al. in 2019 utilized the DAS to assess dementia literacy among community-dwelling adults from Hong Kong [19]. Their study comprised a combined qualitative and quantitative study design with 31 primary care physicians who were interviewed and 437 primary care physicians who completed and returned a questionnaire survey. Their results via factor analysis for both groups confirmed the scale two-factor model, further supporting its validity [19]. The study by Gkioka et al. in 2020, which was conducted with 212 students from the School of Psychology at Aristotle University of Thessaloniki, Greece, also found adequate reliability of the DAS (Cronbach’s α = 0.74) and a two-factor construct validity of social comfort and knowledge as well [18]. Similarly, the study of Teichmann, Melchior, and Kruse in 2022 investigated, via a cross-sectional study, the psychometric properties of the German version of three instruments regarding dementia attitudes, with the DAS among them, in the general population [20]. The study sample comprised 263 people from the general public recruited via an online platform with a mean age of 45.6 years. The results showed excellent internal reliability (α = 0.90), whilst the CFA confirmed a two-factor structure, where all items loaded onto the factors of “comfort” and “knowledge” [20].

On the other hand, the study of Çetinkaya et al., conducted in Turkey in 2020, found a DAS three-factor model [21]. The study attempted to assess the psychometric properties of the Turkish version of the Dementia Attitudes Scale in 326 young and middle-aged students, both from the Faculty of Medicine and the Faculty of Health Sciences at Manisa Celal Bayar University. According to their results, the DAS had a high Cronbach’s alpha coefficient (α = 0.84), whilst CFA revealed a three-factor model comprising (a) “Supporting attitude”, (b) “Accepting attitude”, and (c) “Exclusionary attitude” [21].

In conclusion, the Dementia Attitude Scale has been translated into several languages. Its results proved that it is a crucial instrument for assessing attitudes toward dementia, supported by research on its factor validity and reliability. The scale’s ability to maintain its factor structure across diverse populations and its correlation with real-world experiences make it an essential tool for improving dementia care and education.

Based on the aforementioned information, our study aimed to conduct a confirmatory and exploratory factor analysis of the DAS on a sample of formal and informal caregivers of people with dementia and Behavioural and Psychological Symptoms (BPSD) enrolled in the RECage multicenter clinical trial [22,23]. At this point, it is important to mention that the DAS was administered to the caregiver sample from the RECage clinical trial to assess changes in caregivers’ attitudes resulting from the educational programs implemented in SCU-Bs and to compare these outcomes with those from non-SCU-B settings [23].

Our study was based on the results of the original DAS validation study of O’Connor and McFadden [14]. Therefore, the main hypothesis was that the confirmatory factor analysis applied to the data of the DAS would not reject the hypothesis of a two-factor structural model, which would be summarized under the umbrella of Comfort, as well as knowledge. In addition, given that participants were recruited from six different European countries, the analysis would indicate whether the questionnaire is free from cultural effects and whether it maintains its two-factor structure despite the diversity of the sample.

## 2. Materials and Methods

### 2.1. Design and Procedure

The present study was part of the RECage study, a longitudinal and multicultural study comprising 11 clinical centres from six European countries that started in 2018 and ended in 2022. The main aim of the RECage study was to evaluate the short- and long-term clinical efficacy of the Special Medical Care Units (SCU-Bs) for people with dementia (PwD) and Behavioural and Psychological Symptoms of Dementia (BPSD). SCU-Bs were “residential medical structures lying outside of nursing homes, in general hospitals or elsewhere, e.g., in private, geriatric or psychiatric hospitals, where patients with BPSD are temporarily admitted when their behavioural disturbances are not amenable to control at home” [23]. To this aim, the RECage study compared two cohorts: (a) the PwD treated at the SCU-Bs with the possibility of being admitted for coping with BPSD, and (b) a control group of PwD recruited in centres lacking this facility (non-SCU-B).

The RECage study also had secondary and tertiary objectives, including (a) assessing the quality of life of the patients and their caregivers, (b) evaluating the cost-effectiveness of SCU-Bs and psychotropic drug consumption, (c) determining the SCU-B capacity to delay the time to Nursing Home Placement (NHP), and d) assessing, via the DAS, the change in caregivers’ attitudes resulting from the educational programs implemented in SCU-Bs, as already mentioned in the objectives section [23].

### 2.2. Ethics

#### Protocol Approval and Participants’ Consents

Participants were provided with both oral and written information about the study’s purpose, duration, and scheduled follow-up visits. They were also informed that they could withdraw or refuse participation at any time without affecting their treatment. Informed consent was obtained from all patients and caregivers involved in the study. Competent patients provided written consent themselves; for those unable to do so due to cognitive impairment, consent was obtained from a legal representative or family caregiver, in line with local regulations.

Regarding data protection, participants and caregivers were informed that data would be collected through a web-based electronic case report form (eCRF) developed by Mediolanum Cardio Research, ensuring compliance with FDA 21 CFR part 11 and European regulations concerning security and data protection. Due to the nature of the study, personal data, such as participant names and other identifying information, were handled in accordance with the General Data Protection Regulation (GDPR, 25 May 2018). Sensitive personal data were used exclusively for research purposes, restricted to clinicians involved in the study, and would not be disclosed in any publications or presentations derived from the research.

The study’s protocol was approved by independent ethical committees in the countries and centers that participated in this project and returned to the coordinating center.

### 2.3. Participants

The total sample of the RECage study consisted of 508 caregivers of patients with dementia of any etiology and Behavioral and Psychological Symptoms in Dementia (BPSD). However, the total sample of our study consisted of 485 caregivers, since twenty-three caregivers were excluded because of missing DAS data at baseline. Particularly, 144 (29.7%) were males with a mean age of 63.7 (SD ± 13.6) years, and 341 (70.3%) were females with a significantly (*p* = 0.0108) lower mean age of 60.5 (SD ± 12.0) years. The male caregivers were mainly spouses (50.7%) and sons (43.8%); similarly, the female caregivers were spouses (44.6%) and daughters (48.7%). The difference between the number of years of education of male caregivers (12.6 ± 3.8) and female caregivers (12.1 ± 4.1) was not statistically significant (*p* = 0.2661).

The participants’ caregivers were followed up every 6 months for three years by 11 European clinical centres (6 non-SCU-B and 5 SCU-B) [23]. The data collected regarding the DAS were data from the baseline assessment of the study. Of the 485 corresponding PwD, 272 were females (56.1%), with a mean age of 78.8 years (SD = ±7.53) and a mean education of 7.87 (SD = ±3.95) years. The 213 (43.9%) men had a mean age of 76.7 years (SD = ±8.42) and a mean education of 10.08 (SD = ±4.89) years. The mean score for the Mini-Mental State Examination (MMSE) [24] was 15.5 (SD = ±6.26), whilst the mean score for the Neuropsychiatric Inventory (NPI) [25] was 52.4 (SD = ±19.07).

### 2.4. Tools/Instruments and Methods

#### 2.4.1. The Dementia Attitude Scale

The total scores of the DAS, as well as the results of the two factors addressed in the O’Connor and McFadden validation study [14], were calculated based on the completed items.

#### 2.4.2. The Neuropsychological Battery of Tests Administered in the RECage Study

The battery of tests administered in the RECage study comprised the following: (1) for the assessment of the general cognitive status—the MMSE [24]; (2) for the assessment of the general functional status—the Activities of Daily Living (ADL) scale; (3) for the assessment of neuropsychiatric symptoms—the NPI [25]; (4) for the assessment of the patient’s BPSD—the Cohen–Mansfield Agitation Inventory (CMAI) [26]; (5) for the assessment of the patient’s quality of life—the proxy-version of the Quality of Life–Alzheimer’s Disease (Qol-AD) [27], the EQ-5D-5L scale [28], and the ICECAP-O [29]; (6) for the caregivers’ quality of life—the Adult Carer Quality of life Questionnaire (ACQoL) [30], the EQ-5D-5L [29], and the Caregiver’s Burden Inventory (CBI) [31]; and, finally, for the evaluation of resource utilization and caregiver time—the Resource utilization scale (RUD) [32]. 

### 2.5. Statistical Analysis

Descriptive statistics were calculated for quantitative variables (mean and standard deviation) and qualitative (categorical) variables (absolute and percent frequency) (Table 1).

Confirmatory and exploratory factor analyses (CFA and EFA, respectively) were carried out with PROC CALIS and PROC FACTOR of SAS^®^ Version 9.4, respectively. The CFA hypothesis is that the DAS has a two-factor structure, according to the original validation study of the DAS [14]. The adequacy of fitting was assessed by the χ2 test and the values of the root mean square error of approximation (RMSEA < 0.05 for a good model fit), the standardized root mean square residual (SRMSR < 0.05 for a good model fit), and, finally, the Bentler’s comparative fit index (>0.90 for a good model fit). EFA was carried out, with the aim of obtaining the most parsimonious factor structure with a clear clinical interpretation. Several rotation methods (varimax, promax, and oblimin) were used in EFA after factor extraction with principal component analysis (PCA) to obtain a better differentiation of the factor loadings (SAS/STAT 9.3 User’s Guide. Available online: https://support.sas.com/documentation/onlinedoc/stat/930/ accessed on 25 June 2024).

## 3. Results

### 3.1. Confirmatory Factor Analysis (CFA)

After 10 iterations, convergence was reached. The χ^2^ model fit was 775.721 (df = 169, *p* ≤ 0.0001), statistically rejecting the confirmatory factor model of the DAS. Indeed, the root mean square error of approximation (RMSEA) was 0.0861, greater than the conventional 0.05 value for a good model fit. The standardized root mean square residual (SRMSR) was 0.0781, not close to the conventional 0.05 value for a good model fit. In addition, Bentler’s comparative fit index was 0.7117, much lower than the required value of at least 0.90, leading us to conclude that the model was very poorly fitted. Therefore, considering that each of the four criteria above consistently testifies to an inadequate model fit, it is possible to conclude that the DAS two-factor model proposed in the original validation study of the DAS [14] was not adequate for our data.

### 3.2. Exploratory Factor Analysis (EFA)

The variance explained by each factor before “varimax” rotation was 4.200 (65.1%) for Factor 1, 1.270 (19.7%) for Factor 2, and 0.980 (13.3%) for Factor 3. After “varimax” rotation, the variance explained by each factor was 2.917 (45.2%) for Factor 1, 1.861 (28.8%) for Factor 2, and 1.673 (25.9%) for Factor 3. In the following sections, the DAS items are reported between brackets F1 or F2, according to the original two factors of “Social Comfort” (F1) and dementia knowledge (F2).

According to our analysis, Factor 1 comprises the following nine items: (1) It is rewarding to work with people who have ADRD (F1); (3) People with ADRD can be creative (F1); (4) I feel confident around people with ADRD (F1); (5) I am comfortable touching people with ADRD (F1); (12) It is possible to enjoy interacting with people with ADRD; (13) I feel relaxed around people with ADRD (F1); (14) People with ADRD can enjoy life; (18) I admire the coping skills of people with ADRD (F2); and (19) We can do a lot now to improve the lives of people with ADRD (F2). Therefore, we propose keeping the original name of “Social Comfort”.

Factor 2 comprised the following seven items: (2) I am afraid of people with ADRD (F1); (6) I feel uncomfortable being around people with ADRD (F1); (8) I am not very familiar with ADRD (F1); (9) I would avoid an agitated person with ADRD (F1); (16) I feel frustrated because I do not know how to help people with ADRD (F1); (17) I cannot imagine taking care of someone with ADRD (F1); and (20) Difficult behaviours may be a form of communication for people with ADRD (F2). Therefore, because of the nature of the DAS items loaded on Factor 2, we propose naming this factor “Social Discomfort”.

Finally, Factor 3 is constituted by the following four items: (7) Every person with ADRD has different needs (F1); (10) People with ADRD like having familiar things nearby (F2); (11) It is important to know the history of people with ADRD (F2); and (15) People with ADRD can feel when others are kind to them (F2)”. Therefore, we propose naming Factor 3 “Dementia Knowledge”.

The three-factorial structure of DAS is presented in Table 2. The correlation coefficient between the original DAS Factor 1, “Social Comfort”, and Factor 2, “Knowledge”, is equal to 0.53473 (*p* < 0.0001). Finally, the correlation coefficient between the (a) “new Factor 1” and the “new Factor 2” is equal to 0.46731 (*p* < 0.0001), (b) between the “new Factor 1” and the “new Factor 3” is equal to 0.30357 (*p* < 0.0001), and (c) between the “new Factor 2” and the “new Factor 3” is equal to 0.24574 (*p* < 0.0001). Similar values have been obtained for the non-parametric Spearman’s correlation coefficient.

Since O’Connor and McFadden [14] derived their two-factor structure of the DAS using a factor analysis with oblimin rotation in 2010 (Table 1 on page 6, with a misplacement of item (3), People with ADRD can be creative, as it should have been categorized with the second set of items), we also repeated our EFA with this kind of rotation to check whether the different factor structure could be due to our varimax rotation. However, oblimin rotation confirmed our varimax results, except for item 20, “Difficult behaviours may be a form of communication for people with ADRD”, which now has a greater load (0.37392) on Factor 1 (social comfort) than on Factor 2 (social discomfort), with a load of 0.22025. The load of this item on Factor 1 is similar to the load (0.34585) on Factor 3 (knowledge).

## 4. Discussion

The DAS is considered to be a valid and reliable tool translated into several languages, which is utilized for assessing attitudes toward dementia and for investigating whether the usual attitudes regarding dementia can change as a result of psychoeducation. For example, the results of the RECage study, in which DAS was used for investigating caregivers’ attitudes regarding dementia, showed that the caregivers of patients admitted to SCU-Bs, in comparison to the caregivers of people with dementia treated at centres lacking a SCU-B, had an improved attitude toward dementia, as assessed using the DAS, due to the psychoeducation and psychotherapy interventions received in SCU-Bs [22]. Since the attitude towards dementia could be a useful outcome in the RECage multicultural study [23], the DAS was included as one of the assessment scales. The present study aimed to conduct a validation analysis of this scale and a confirmatory factor analysis of the original factor structure shown in the paper of O’Connor and McFadden, in 2010 [14], as well as an explanatory factor analysis.

We did not confirm the two-factor structure of the DAS presented by O’Connor and McFadden (2010) [14] and other studies that support the two-factorial model of DAS [19,20,21], supporting the idea that the DAS could maintain its factor structure across diverse populations.

It is important to stress that O’Connor and McFadden conducted their factor analysis on two different samples of students: college students (n = 302) and certified nursing assistant students (n = 145) [14]. Furthermore, the confirmatory factor analysis was then conducted with another sample of college students (n = 157). It is very evident that their results cannot be generalized to a population of spouses, sons, or daughters who are usually the caregivers of people with dementia, as in our RECage study, which, from our point of view, is the common occurrence in the majority of cases.

On the contrary, our results indicated a three-factor structure, with nine DAS items loading onto the “Social Comfort” factor, capturing positive attitudes towards individuals with dementia; seven items loading onto the new factor of “Social Discomfort”; and the last four items loading onto the “Dementia Knowledge” factor. Our current findings propose that the two-factorial model of the previous studies that applied the DAS may oversimplify the complexities of attitudes toward individuals with dementia. According to the literature, only the study by Çetinkaya et al. in 2020 indicated a three-factor structure model comprising the “Supporting attitude”, the “Accepting attitude”, and the “Exclusionary attitude” [21]. When comparing the results of the current study with those of Çetinkaya et al., it is noteworthy that both studies underscore the importance of addressing emotional and social factors in dementia education and training. Çetinkaya et al. found that negative attitudes toward dementia were prevalent among their sample, which echoes the findings of the current study regarding the “Social Discomfort” factor [21]. However, it has to be pointed out that the three-factor structure shown in the paper of Çetinkaya et al. has some aspects that were difficult to interpret. For example, the DAS item 16 -I feel frustrated because I do not know how to help people with ADRD- that loaded onto Factor 1, i.e., the Supportive attitude, seems to fit better to factor 3 which was defined as “Exclusionary attitude”. Similarly, other items, such as item 3—People with ADRD can be creative—and item 14—People with ADRD can enjoy life—which were assigned to Factor 1, seem to fit well in both Factor 1 and Factor 2. Furthermore, the classification of the study by Çetinkaya et al. (2020) into “Supporting”, “Accepting”, and “Exclusionary” attitudes, besides the fact that it attempts to capture the multifaceted nature of attitudes toward dementia, seems not to be aligned with the existing literature on dementia attitudes [21]. From a psychological perspective, it is worth mentioning that the aforementioned categorization of attitudes may oversimplify the complex emotional and cognitive dimensions of attitudes toward dementia, since they are influenced by various factors, including knowledge and education about the condition [33,34]. Moreover, the term “Exclusionary Attitude” may not fully encapsulate the reasons behind negative attitudes towards dementia. Since attitudes seem to be influenced by social norms and beliefs, this can lead to exclusionary behaviours not necessarily rooted in malice but rather in a lack of understanding or fear [35]. Therefore, the study’s classification of dementia attitudes into “Supporting”, “Accepting”, and “Exclusionary” may oversimplify the underlying dynamics and the influence of knowledge, education, and behavioural discrepancies.

Our three-factor model possibly reflects that general knowledge about dementia is not merely a cognitive construct but is intertwined with emotional and social dimensions of comfort and discomfort in interactions with individuals with dementia. These results may highlight the complexity of attitudes towards individuals with dementia and the need for a more comprehensive understanding of these constructs [13,36], suggesting a consistent pattern across different cultural contexts where knowledge alone may not mitigate negative attitudes and emphasizing the need for comprehensive educational interventions that address both knowledge and emotional responses. Furthermore, the current results regarding the three-factorial model suggest that training programs for healthcare professionals should not only focus on enhancing knowledge about dementia but also on fostering positive attitudes and comfort in interacting with individuals with dementia. This dual approach may be crucial in improving the quality of care provided to this population [19,37].

According to EFA, the variance explained by the factors before and after varimax rotation indicated a significant reorganization of the items and a three-factor structure comprising “Social Comfort”, “Social Discomfort”, and “Dementia Knowledge”, with Factor 1 capturing positive attitudes towards individuals with dementia and Factor 2 reflecting negative feelings and discomfort associated with dementia care. On the other hand, Factor 3 encompassed general knowledge about dementia. As it seems, EFA provides a more nuanced understanding of how attitudes towards dementia are organized. The correlation coefficients among the newly defined factors in our study suggest a moderate relationship between social comfort and discomfort and a weaker correlation between social comfort and general knowledge, indicating that while there is some overlap in attitudes, they are distinct constructs. This finding aligns with previous studies that have shown that knowledge about dementia does not necessarily correlate with positive attitudes towards individuals living with dementia, emphasizing the importance of addressing both knowledge and attitudes in educational interventions [13,38,39]

Finally, as regards item 20, “Difficult behaviors may be a form of communication for people with ADRD”, it loaded onto Factor 2 after varimax rotation. However, its classification is not definitive, as it could also fit Factor 1, “Social Comfort”, following the oblimin rotation. This discrepancy probably arises due to the caregivers’ difficulty in interpreting the item. We hypothesize that the question is mainly aimed at assessing the caregiver’s understanding of ADRD but that it could also be understood as addressing whether the caregiver feels uncomfortable when confronted with difficult behaviours exhibited by individuals with ADRD. In our opinion, this poorly defined question should be removed or rather rewritten for a better understanding. However, for the time being, this item shall be classified into Factor 1 as an expression of a form of comfort for caregivers dealing with these patients.

### 4.1. Strengths of the Study

The results of the study regarding the DAS have the notable strengths of being a multicultural representation of the sample and of being obtained with a substantial sample size. The inclusion of participants from six European countries enhances the generalizability of the findings and allows a more comprehensive understanding of attitudes toward dementia across diverse cultural contexts. By incorporating a multicultural sample, the current study aligns with the growing recognition of the importance of cultural contexts in dementia care and attitudes. Moreover, the large sample size of 485 participants’ caregivers strengthens the reliability and validity of the study findings. The large sample size increases the statistical power of our analysis, allowing for more robust conclusions to be drawn from our data.

### 4.2. Limitations and Future Work

Every study has its limitations, and this one is no exception. A fundamental limitation of this study stems from a common challenge encountered in all research focused on identifying factors within rating scales, including psychiatric, medical, and psychological scales. Specifically, the structure of any given scale is often influenced by the specific case series to which the rating scale is applied, which in turn limits the generalizability of findings from studies using confirmatory factor analysis (CFA). Consequently, we advise future researchers to be mindful of this limitation. Additionally, it is crucial for each study to independently assess the scale structure for their particular sample and patient population.

## 5. Conclusions

In conclusion, our study supports the idea of a three-factor structure of the DAS, which comprises (a) social comfort, (b) social discomfort, and (c) dementia knowledge. This finding challenges the adequacy of the original study and the vast majority of the other studies that support a two-factor model of the DAS. Therefore, our results suggest a more complex three-factor structure that possibly better captures the nuances of attitudes toward dementia. Furthermore, it has to be noted that the three-structure model of the DAS was determined through a significantly diverse sample of caregivers from six different European countries. Consequently, we consider that further studies including different cultures have to be conducted to investigate cultural effects on the DAS.

In comparison to other research in the literature, our study reinforces the notion that attitudes toward dementia are multifaceted and influenced by both knowledge and emotional factors. Future research should continue to explore these dimensions and consider implications for educational interventions aimed at improving attitudes and care for people with dementia.

## Figures and Tables

**Table 1 neurosci-06-00045-t001:** Descriptive statistics of the 20 items of the DAS along with the two-factor total scores, according to the original study and the total scores in each cohort (SCU-B and non-SCU-B) and overall.

DAS Items	SCU-B (n = 248)		non SCU-B (n = 237)		Total (n = 485)	
	Mean (SD)	Median (Q1–Q3)	Mean (SD)	Median (Q1–Q3)	Mean (SD)	Median (Q1–Q3)
1. It is rewarding to work with people who have ADRD	3.73 (1.74)	4.0 (2.0–5.0)	3.63 (1.80)	4.0 (2.0–5.0)	3.68 (1.77)	4.0–(2.0–5.0)
2. I am afraid of people with ADRD	5.38 (1.77)	6.0 (4.0–7.0)	5.41 (1.85)	6.0 (4.0–7.0)	5.39 (1.80)	6.0–(4.0–7.0)
4. I feel confident around people with ADRD	3.80 (1.84)	4.0 (2.0–5.0)	3.72 (1.82)	4.0 (2.0–5.0)	3.76 (1.83)	4.0–(2.0–5.0)
5. I am comfortable touching people with ADRD	4.60 (1.85)	5.0 (4.0–6.0)	4.85 (1.81)	5.0 (4.0–6.0)	4.72 (1.83)	5.0–(4.0–6.0)
6. I feel uncomfortable being around people with ADRD	4.88 (1.88)	5.0 (4.0–7.0)	5.27 (1.77)	6.0 (4.0–7.0)	5.07 (1.84)	5.0–(4.0–7.0)
8. I am not very familiar with ADRD	4.13 (2.17)	4.0 (2.0–6.0)	3.93 (2.11)	4.0 (2.0–6.0)	4.04 (2.14)	4.0–(2.0–6.0)
9. I would avoid an agitated person with ADRD	4.81 (1.91)	5.0 (3.0–7.0)	4.70 (1.96)	5.0 (3.0–7.0)	4.75 (1.93)	5.0–(3.0–7.0)
13. I feel relaxed around people with ADRD	3.44 (1.68)	3.0 (2.0–5.0)	3.59 (1.75)	4.0 (2.0–5.0)	3.51 (1.71)	4.0–(2.0–5.0)
16. I feel frustrated because I do not know how to help people with ADRD	3.66 (1.77)	3.0 (2.0–5.0)	3.74 (1.93)	3.0 (2.0–5.0)	3.70 (1.85)	3.0–(2.0–5.0)
17. I cannot imagine taking care of someone with ADRD	5.22 (1.88)	6.0 (4.0–7.0)	5.19 (1.82)	6.0 (4.0–7.0)	5.20 (1.85)	6.0–(4.0–7.0)
3. People with ADRD can be creative	4.21 (1.63)	4.0 (3.0–5.0)	3.98 (1.77)	4.0 (2.0–5.0)	4.10 (1.70)	4.0–(3.0–5.0)
7. Every person with ADRD has different needs	6.09 (1.12)	6.0 (6.0–7.0)	6.22 (1.09)	7.0 (6.0–7.0)	6.16 (1.11)	6.0–(6.0–7.0)
10. People with ADRD like having familiar things nearby	5.96 (1.24)	6.0 (5.0–7.0)	6.16 (1.26)	7.0 (6.0–7.0)	6.08 (1.25)	6.0–(6.0–7.0)
11. It is important to know the past history of people with ADRD	5.85 (1.28)	6.0 (5.0–7.0)	6.16 (1.15)	6.0 (6.0–7.0)	6.00 (1.23)	6.0–(6.0–7.0)
12. It is possible to enjoy interacting with people with ADRD	4.84 (1.51)	5.0 (4.0–6.0)	4.46 (1.73)	5.0 (4.0–6.0)	4.66 (1.63)	5.0–(4.0–6.0)
14. People with ADRD can enjoy life	4.11 (1.89)	4.0 (3.0–6.0)	3.88 (1.94)	4.0 (2.0–5.0)	4.00 (1.92)	4.0–(2.0–6.0)
15. People with ADRD can feel when others are kind to them	6.00 (1.14)	6.0 (6.0–7.0)	5.81 (1.47)	6.0 (5.0–7.0)	5.91 (1.31)	6.0–(5.0–7.0)
18. I admire the coping skills of people with ADRD	4.06 (1.63)	4.0 (3.0–5.0)	4.47 (1.74)	4.0 (4.0–6.0)	4.26 (1.70)	4.0–(3.0–6.0)
19. We can do a lot now to improve the lives of people with ADRD	5.15 (1.60)	6.0 (4.0–6.0)	5.61 (1.39)	6.0 (5.0–7.0)	5.38 (1.52)	6.0–(5.0–6.0)
20. Difficult behaviors may be a form of communication for people with ADRD	5.29 (1.44)	6.0 (4.5–6.0)	5.00 (1.53)	5.0 (4.0–6.0)	5.15 (1.49)	5.0–(4.0–6.0)
DAS—Factor 1. Comfort-filled items	43.63 (9.71)	43.0 (37.0–51.0)	44.03 (11.33)	45.0 (37.0–52.0)	43.83 (10.52)	44.0–(37.0–51.0)
DAS—Factor 2. Knowledge-filled items	51.61 (8.18)	52.0 (46.0–57.0)	51.76 (8.45)	52.0 (48.0–57.0)	51.68 (8.31)	52.0–(47.0–57.0)
Total DAS completed items	95.24 (15.80)	96.0 (84.0–106.0)	95.79 (16.95)	97.0 (86.0–108.0)	95.51 (16.36)	97.0–(85.0–107.0)

Notes: The ten items in italics constitute the first factor (social comfort), while the remaining ten items constitute the second factor (knowledge). Values are calculated at the baseline on all the completed items. Abbreviations: SCU-B: Special Medical Care Unit for people with dementia; non SCU-B: clinical centres lacking SCU-B facilities; ADRD: Alzheimer’s disease and related dementias; DAS: Dementia Attitude Scale.

**Table 2 neurosci-06-00045-t002:** Factor analysis on 20 DAS items after varimax rotation.

DAS Items	Factor 1 (Comfort)	Factor 2 (Discomfort)	Factor 3 (Knowledge)
1. It is rewarding to work with people who have ADRD	* 0.61459	0.22792	0.07950
2. I am afraid of people with ADRD	0.09875	* 0.56877	0.07798
3. People with ADRD can be creative	* 0.48319	0.11173	0.16315
4. I feel confident around people with ADRD	* 0.52742	0.24881	−0.06057
5. I am comfortable touching people with ADRD	* 0.39847	0.33945	0.12806
6. I feel uncomfortable being around people with ADRD	0.23548	* 0.57335	0.02266
7. Every person with ADRD has different needs	−0.01834	0.11068	* 0.47035
8. I am not very familiar with ADRD	−0.00465	* 0.39354	0.17037
9. I would avoid an agitated person with ADRD	0.15390	* 0.60291	0.04428
10. People with ADRD like having familiar things nearby	0.08056	0.05709	* 0.66414
11. It is important to know the past history of people with ADRD	0.20755	0.07553	* 0.52764
12. It is possible to enjoy interacting with people with ADRD	* 0.55411	0.22151	0.24083
13. I feel relaxed around people with ADRD	* 0.67792	0.26591	−0.10025
14. People with ADRD can enjoy life	* 0.51657	0.03625	0.14412
15. People with ADRD can feel when others are kind to them	0.13901	0.09688	* 0.55877
16. I feel frustrated because I do not know how to help people with ADRD	0.29828	* 0.33589	−0.03068
17. I cannot imagine taking care of someone with ADRD	0.12667	* 0.40925	0.17912
18. I admire the coping skills of people with ADRD	* 0.52845	0.01975	0.13990
19. We can do a lot now to improve the lives of people with ADRD	* 0.44447	0.01232	0.29688
20. Difficult behaviours may be a form of communication for people with ADRD	−0.30865	* 0.12810	−0.31530

* The greatest factor loading of the items on each factor.

## Data Availability

Data are available from Bruno Cesana and Carlo Alberto Defanti.

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
