# Peer review of "A Confirmatory Factor Analysis of the Dementia Attitude Scale (DAS) in a European Case Series of Caregivers of People with Dementia Enrolled in the RECage Study"

_neurosci, 2025, doi:10.3390/neurosci6020045_

Round 1
Reviewer 1 Report
Comments and Suggestions for Authors
Very Respected Authors,
After carefully reading your paper I have few suggestions.
Abstract is well-written. The title of the paper is too long. The objective of the paper is usually not separated as a separate sub-chapter. I is usually at the end of the introduction. Approval from the Ethics committee is usually in the methodology section. Adequate statistical analysis was applied. Results are clearly presented. The conclusion have to be in the agreement with the objective and findings, not general. The used reference are appropriate.
1.
The main question addressed by the research seems to be focused on understanding the attitudes of caregivers towards dementia, particularly in cases where individuals with dementia exhibit behavioral and psychological problems.
Yes, it is an original paper. The research likely aims to explore how caregivers perceive and respond to the challenges posed by caring for individuals with these specific symptoms of dementia. Additionally, it seems to involve analyzing a specific scale (DAS) to measure these attitudes, using data from a European study of caregivers.
The research could also be investigating how these attitudes are influenced by factors such as the severity of the behavioral and psychological symptoms of dementia, as well as the broader context of caregiving in European countries. Confirmatory factor analysis" (CFA) is a statistical technique used to test and confirm the factor structure in a scale or questionnaire. In this context, CFA was used to confirm the structure of the scale that measures attitudes or perceptions regarding dementia. This means that researchers use CFA to test whether the data they collected from respondents (in this case, caregivers of individuals with dementia) align with the expected structure of the scale.
The scale probably has several dimensions that allow for the measurement of different aspects of attitudes toward dementia, such as empathy, fear, stigmatization, or promoting social inclusion. "DAS" is likely an acronym that stands for the name of this scale. It could be a specific name developed for the purposes of the research and is used as an abbreviation in this context.
The part - scale 2. To dementia (DAS)" refers to a specific scale that measures attitudes toward dementia. The scale is likely designed to assess how prone people are to negative or positive attitudes toward individuals with dementia.
In a European series of cases of caregivers of individuals with dementia - This part of the title refers to a study conducted on a sample of caregivers who care for individuals with dementia, within the context of European countries.
2.
This paper may contribute new insights into the attitudes and behaviors of caregivers toward individuals with dementia in the context of European research, using specific methods of analysis. It may also provide a better understanding of how specific factors (such as behavioral and psychological problems) influence caregivers' attitudes. When compared to previous research, this paper may contribute by initiating further studies within the European context, using methodological approaches that may have been underutilized or specific to this field. The use of specific scales or methods, such as Confirmatory Factor Analysis (CFA), allows for a deeper understanding or validation of the structure of scales used to measure attitudes toward dementia.
3. The study not only expands existing knowledge but also opens the door for future studies that can further refine and validate tools for assessing attitudes toward dementia, ultimately helping to improve the quality of care for patients with dementia and better support for their caregivers in Europe.
Comments on the Quality of English LanguageVery Respected Editor,
Accept this manuscript for publication after after correction.
Reviewer 2 Report
Comments and Suggestions for Authors
Thank you for opportunity to review referenced manuscript on the important topic of dementia. The authors did a great job in the writing of the manuscript.
I found some areas that were confusing to the reader, such as the description of the methods. It indicates that the study is based on a previous study where the study participants involved spouses, sons daughters etc. the authors conclusion indicates differently and refers as healthcare workers at the facilities where the study took place.
Also recommend discussing limitations of the study and future recommendations.
Round 2
Reviewer 1 Report
Comments and Suggestions for Authors
Both the abstract and the introduction are well written. The aim of the paper is clear and is no longer presented as a separate subsection. Lines 176 to 181 should also be included in the aim of the study as additional objectives, rather than in the subsection Material and Methods. The methodology is described in detail. The results are clearly presented. The discussion is well written, as is the conclusion.
Comments on the Quality of English LanguageThe authors have revised the most important parts of the paper. Lines 176 to 181 should also be included in the aim of the study as additional objectives, rather than in the subsection Material and Methods.
Accept for publication after minor revision.
